# Therapeutic Applications of the CRISPR-Cas System

**DOI:** 10.3390/bioengineering9090477

**Published:** 2022-09-15

**Authors:** Kyungmin Kang, Youngjae Song, Inho Kim, Tae-Jung Kim

**Affiliations:** 1College of Medicine, The Catholic University of Korea, 222 Banpo-daero, Seocho-gu, Seoul 06591, Korea; 2Department of Hospital Pathology, Yeouido St. Mary’s Hospital, College of Medicine, The Catholic University of Korea, 10, 63-ro, Yeongdeungpo-gu, Seoul 07345, Korea

**Keywords:** CRISPR-Cas9, Cas13, Immunotherapy, gene editing, cell therapy

## Abstract

The clustered regularly interspaced palindromic repeat (CRISPR)-Cas system has revolutionized genetic engineering due to its simplicity, stability, and precision since its discovery. This technology is utilized in a variety of fields, from basic research in medicine and biology to medical diagnosis and treatment, and its potential is unbounded as new methods are developed. The review focused on medical applications and discussed the most recent treatment trends and limitations, with an emphasis on CRISPR-based therapeutics for infectious disease, oncology, and genetic disease, as well as CRISPR-based diagnostics, screening, immunotherapy, and cell therapy. Given its promising results, the successful implementation of the CRISPR-Cas system in clinical practice will require further investigation into its therapeutic applications.

## 1. Introduction

CRISPRs (clusters of regularly inter-spaced palindromic repeats) is a term that refers to regular and repetitive nucleotide sequences that are universally present in bacterial DNA. It was first discovered in 1987 by Professor Nakada’s team at Osaka University in Japan [1]. Since then, several researchers have discovered that this repeating sequence is the microorganism’s defense system for fighting virus invasion [2]. That is, when a microorganism is infected with a bacteria-specific virus, some of the surviving individuals cut part of the virus’s DNA and insert it into their genome one after another. Bacteria defend themselves against phages and plasmids by using the CRISPR-CRISPR-associated protein (Cas) system, which is a genetically encoded RNA-mediated adaptive immunity system. The CRISPR-Cas system is extremely diverse, and it is broadly classified into two classes (I and II), which are further classified into six types (I-VI) and several subtypes [3,4]. Cascade is the effector complex for CRISPR interference in class I/type I systems. It is composed of CRISPR RNA (crRNA), a large and small Cas subunit, as well as a variable number of Cas5, Cas6, and Cas7 that recognize the target and recruit Cas3, cleaving the target DNA [5,6,7]. On the other hand, the RNP surveillance complex in class II systems is made up of a single Cas protein, Cas9 or Cpf1, and this effector complex participates in both target recognition and cleavage [5,8].

RNAs are transcribed from the CRISPR locus and processed to produce crRNA and sequence-invariant trans-activating crRNA (tracrRNA), with a nucleotide sequence corresponding to the spacer [9]. These RNAs combine with a protein called Cas9 (CRISPR-associated protein 9) to form a sequence-specific endonuclease [10]. Using this principle, the possibility of gene-editing technology using RNA has been reported, unlike the previously used ZFN and TALEN gene-editing technologies. For gene editing using CRISPR, single-guide RNA (sgRNA) can be made by linking the major regions of crRNA and tracrRNA [11]. With the Cas9 system derived from S. pyogenes, it was demonstrated that the Cas9-sgRNA complex can cleave target DNA with a nucleotide sequence complementary to that of sgRNA in vitro [12].

Following the discovery of the possibility of gene editing using the CRISPR-Cas system, numerous studies and experiments were carried out in various laboratories. Because of its convenience of use and the technology’s strength, it has a broad impact, not just in the field of basic biology but also in medical applications [10]. In addition to gene editing with Cas9, which received early attention, diverse advancements in methods such as cas12a and cas13 are being produced, and which are widely used in gene-based biomedical engineering [13]. Gene editing in medicine, in particular, can sometimes suggest ideas of solving difficulties with current methods of treatment, and it can also provide a long-term solution for genetic diseases through fundamental gene editing [14]. In this review article, the current medical treatment with the CRISPR system is divided into three categories: genetic diseases, cancer diseases, and infectious diseases.

## 2. Therapeutic Approach for Genetic Disease

Genetic disease occurs when a specific mutation of a parent’s gene is inherited and does not function normally [15]. When a genetic disease developed symptoms, it was possible to avoid the substances that cause it or to provide symptomatic treatment to alleviate the symptoms, but the underlying treatment method has yet to be widely adopted [16]. To overcome this, and for a more fundamental treatment, researchers use gene-editing technology, i.e., the CRISPR-Cas9 system, to treat the defective gene [14,17,18]. The CRISPR-Cas9 system is used in treatment to correct genes such as blood and somatic cells taken from the patient’s body in a laboratory and then injected back into the patient’s body [19]. When this treatment is repeated several times, the percentage of normal cells in the body increases, which can completely cure the disease or relieve symptoms [20]. In addition, a method of injecting a genetically modified material tailored to the patient’s condition into the human body is used. When the CRISPR-Cas9 system enters the body, it comes into contact with somatic cells, which precisely locate the target DNA and begin gene editing [21]. So now, the CRISPR–Cas system is now widely used to treat human genetic diseases, such as cystic fibrosis (CF), Duchenne muscular dystrophy (DMD), Huntington’s disease (HD), hemophilia, and hematopoietic diseases.

### 2.1. Cystic Fibrosis

Cystic fibrosis (CF) is a disease caused by a defect in the CTFR proteins, which affects several organs and causes a loss of control over electrolyte and osmole because mucus is not created normally [22]. It was thought to be an excellent candidate for a genetic engineering-based treatment approach. Since it was discovered that utilizing CRISPR-cas9 in iPSCs extracted from CF patients, it is feasible to efficiently fix the homozygous deletion of F508 in the CFTR gene, which is the major mutation of disease [23]. Since then, methods for accurately and safely differentiating ex vivo edited cells by injecting them back into the body are being studied [24,25]. Furthermore, research is being performed to circumvent the limits of existing approaches for correcting genes by targeting epithelial or basal cells directly. Novel approaches are also being developed continuously, such as the double nickase approach and numerous permutations of size and symmetry in repair template homology arms [26].

### 2.2. Duchenne Muscular Dystrophy

Duchenne muscular dystrophy (DMD) is the most common hereditary disease among progressive muscular dystrophy and is caused by a dystrophin gene mutation [27]. Dystrophin is part of a group of proteins (a protein complex) that work together to strengthen and protect muscle fibers from injury [28]. As a result, in DMD patients, gradual degeneration occurs primarily in skeletal muscle, with connective tissue or fat replacing the muscle, resulting in pseudohypertrophy and decreased muscle strength. Exon 50 of dystrophin’s rod domain is one of the most common deletions in DMD patients, putting exon 51 out of frame with preceding exons [29]. Previous research has shown that exon 51 can be skipped or reframed, and dystrophin expression can be restored by injecting two adeno-associated viruses of serotype 9 (AAV9) which encode the CRISPR-Cas9 gene and sgRNAs into a canine model [30]. Another study proposed two strategies for correcting this mutation by CRISPR-Cas9-mediated skipping of surrounding exons, causing splicing of exon 43 to exon 45, and introducing a premature termination codon in exon 44 of the dystrophin gene [31].

### 2.3. Huntington’s Disease

Huntington’s disease (HD) is a neurodegenerative disorder that is caused by a CAG trinucleotide repeat expansion in exon 1 of the huntingtin (HTT) gene, which results in the production of abnormal proteins that gradually damage brain cells [32]. Using SaCas9-induced indels, researchers used the CRISPR-Cas9 system to disrupt the expression of the mutant HTT gene in a mouse model of HD, resulting in a nearly 50% reduction in neuronal inclusions, as well as a significant improvement in life span and some motor impairments [33].

### 2.4. Hemophilia

Hemophilia is a congenital hemorrhagic disease caused by mutations in the blood coagulation genes factor VIII (FVIII) or factor IX (FIX) [34]. Hemophilia is a promising target for gene therapy because it is caused by a single genetic defect. As a result, many researchers are attempting experiments with the CRISPR-Cas9 system to correct the defective coagulation factor gene [35], and many clinical trials for hemophilia A and hemophilia B are currently underway [36]. Nonetheless, many limitations remain, such as the immune response to the AAV used to deliver the CRISPR-Cas9 system or tracking the long-term effect after treatment, and various methods for circumventing these issues are currently being proposed [37].

### 2.5. Hemoglobinopathy

Hemoglobinopathies, which include β-thalassemia and sickle cell disease (SCD), are genetic diseases caused by disorder in the proteins that form the structure of hemoglobin, which transports oxygen [38]. The study was carried out in patients with these diseases under the assumption that if fetal hemoglobin was reactivated, clinical symptoms in patients with reduced oxygen transport capacity due to abnormal hemoglobin would be mild [39]. Researchers developed the CRISPR-Cas9 genome-editing strategy in KU-812, KG-1, and K562 cell lines by deleting a 200 bp genomic region within the human erythroid-specific BCL11A (B-cell lymphoma/leukemia 11A) enhancer. The deletion of 200 bp of the BCL11A erythroid enhancer, which includes the GATAA motif, results in a significant increase in γ-hemoglobin expression in K562 cells [40].

## 3. Cancer Therapeutics

Cancer is a genetic disease caused by a series of genetic/epigenetic errors [41]. The development of the CRISPR-Cas9 system suggested that researchers might be given a new method for cancer treatment [42]. Given that cancer is a genetic abnormality caused by a series of genetic alterations, it is reasonable to believe that repairing oncogenic genome/epigenome aberrations using CRISPR-Cas9 could be a promising cancer treatment modality [43]. Furthermore, by using the CRISPR system, existing anticancer therapies can be improved in terms of high precision and safety, and broad applications including anticancer drug development research are possible [44].

### 3.1. Genome Editing of Cancer Cells

Because of the specificity, efficiency, and accuracy of CRISPR-Cas9, this genome editing tool is widely used in research laboratories, allowing researchers to identify the role of various oncogenes in cancer cells [45]. Applications of CRSIPR-Cas9 for cancer gene edition (Figure 1) and subsequent cancer therapeutics (Figure 2) are demonstrated.

#### 3.1.1. Correction of Tumor Suppressor Genes

Multiple genes in the protein kinase C (PKC) gene family regulate cellular functions such as cell survival and proliferation [46], and also act as a tumor suppressor [47]. The correction of a loss-of-function PKC mutation in a patient-derived colon cancer cell line using CRISPR-cas9 genome editing inhibited by anchorage-independent growth and reduced tumor progression in a xenograft model [48]. The phosphatase and tensin homolog (PTEN) gene acts as a tumor suppressor gene through the action of its phosphatase protein product, and the PTEN gene mutation causes tumorigenesis. Researchers used the CRISPR system, specifically Cas9 fused to the transactivator VP64-p65-Rta (VPR), to activate PTEN expression in melanoma and TNBC cell lines. The PTEN protein is produced when the PTEN gene is corrected normally, and it inhibits downstream oncogenic pathways such as AKT, mTOR, and MAPK signaling [49].

#### 3.1.2. Inhibition of Proto-Oncogenes

The epidermal growth factor receptor (EGFR), a major driver of cellular proliferation, differentiation, migration, and angiogenesis, plays an important role in the progression of lung cancer [50]. CRISPR-Cas9 can be used to knock out the oncogenic mutant EGFR allele, which prevents the growth and proliferation of lung cancer cell lines and reduces tumor volumes in xenograft mice [51]. δ-Catenin, which is encoded by the CTNND2 gene, has been found to be overexpressed in a variety of cancers. The researchers discovered that when δ-catenin proteins were depleted in vivo by knocking out the CTNND2 gene using CRISPR-Cas9 technology, tumorigenesis of Lewis lung cells and stromal effects were reduced. Such research suggests that the discovery of an oncogene may lead to the identification of new lung cancer therapeutic targets [52]. CD133 is the most commonly used marker for cancer stem cell isolation, and CD133 expression has been linked to poor prognosis, metastasis, and recurrence in colon cancer [53]. The CRISPR-Cas system effectively reduced cell proliferation and colony formation in colon cancer cells by knocking out CD133, demonstrating the effect of vimentin loss [54].

#### 3.1.3. Dysregulation of Chemoresistance-Related Genes

If the nuclear factor erythroid 2-related factor (NRF2) gene is overexpressed, tumor cells create an environment that is resistant to anticancer drugs, lowering the effectiveness of current chemotherapy drugs [55]. Then, experiments showed that when the NRF2 gene was removed from lung cancer cells in A549 cells using the CRISPR-Cas system, the degree of response to chemotherapy, such as cisplatin and carboplatin, was more sensitive than non-knockout cells [56]. Through the NF-κB pathway, RSF-1 induces paclitaxel resistance in cancer cells. RSF-1 knockout resulted in G1 phase cell arrest, increased cell apoptosis, and decreased cell migration and proliferation in H460 and H1299 cells. In H460 cell xenograft mice, Rsf-1 knockout enhanced the paclitaxel-mediated reduction in tumor volume and weight [57].

### 3.2. Cancer Diagnosis and Therapy

#### 3.2.1. CRISPR-Based Diagnostics

Even though a number of cancer detection methods are frequently employed, they still need to be enhanced in terms of sensitivity and specificity. It is essential to identify sensitive genes through genetic diagnosis to prevent cancer [58]. The highly sensitive CRISPR-based mutation detection of BRAF V600E and EGFR L858R was introduced [59]. The detection of high-risk HPV types in tissue samples [60], such as HPV16 and HPV18, which are important diagnostics in the current treatment for cervical cancer [61] and head and neck cancer [62], is described. Recently, programmable DNA-binding probes, such as fluorescent CRISPR emerged as powerful tools for imaging cancer diagnostics, which facilitate the visualization of chromatin dynamics and nuclear architecture [63].

#### 3.2.2. CRISPR Screening

Functional genomic screening using CRISPR-Cas9 has demonstrated promise as an objective technique for locating unidentified cancer targets [64]. Novel immuno-oncology targets and tumor immune modulators have also been found using this technique, and their underlying mechanisms have been examined. Multiple regulators of PD-L1 and/or major histocompatibility complex (MHC) class I have been discovered using CRISPR screens, potentially facilitating combination immunotherapies for cancer [65,66]. In vivo CRISPR screens have been shown to be effective in identifying such cancer-cell-intrinsic tumor microenvironment regulators, which are associated with heterogeneous cytokine distributions and may influence cell-extrinsic effects [67]. In melanoma cells, the CRISPR-Cas9 knockout library was used to look for novel candidate genes whose deletion conferred resistance to vemurafenib, a BRAF protein kinase inhibitor [68], and for the genetic susceptibility of acute myeloid leukemia [69].

#### 3.2.3. PD1/PDL1 Immunotherapy

PD-L1 is found in a wide range of immune and cancer cells. By interacting with PD-1 on T cells, PD-L1 inhibits T cell activity and growth, promotes T cell exhaustion, and induces apoptosis in activated T cells, allowing tumor cells to evade host immunity [70]. As a result, blocking the interaction between PD-1 and PD-L1 with inhibitors allows T cells to normally kill and eliminate tumor cells, a treatment strategy that is effective for anti-tumor immunity [71]. The precise knockout of the PD-1 gene successfully reduced PD-1 expression via the electroporation of plasmid-encoded sgRNA and Cas9 into human T cells, enhancing T cell immune responses to cancer cells and the ability to effectively kill cancer cells [72]. CRISPR-edited T cells with a knockout PD-1 gene were used in the first human trial to treat patients with metastatic non-small cell lung cancer who had failed to respond to chemotherapy, radiation, and other therapies [73,74]. Additionally, clinical trials of CRISPR-mediated PD-1 gene knockout are being conducted in patients with prostate cancer, bladder cancer, hepatocellular carcinoma, advanced esophageal cancer, and metastatic renal cell carcinoma [75].

#### 3.2.4. Cell Therapy

Cell therapy manipulates immune cells in vitro through genetic engineering and then administers these cells to patients in order to combat cancer [76]. Previous research has shown tremendous success in hematologic malignancies [77]. Chimeric antigen receptors (CARs) are synthetic receptors that allow T cells to recognize tumor-associated antigens in an MHC-independent manner [78], acting as a transmembrane domain, a hinge segment, an antibody-derived extracellular-specific target protein-binding domain, and a T-cell-activating intracellular signaling unit [79]. CARs are inserted into the autologous T cells that are collected from the patients, resulting in the expression of CARs on the surface of T cells [80]. The CRISPR-Cas9 system on CAR-T cells for solid tumors can disrupt multiple genomic sites at the same time, resulting in universal CAR-T cells lacking endogenous TCR, HLA class I (HLA-I), and PD-1 [81]. CRISPR-Cas9 technology has recently been used to create more effective CAR-T cells through the targeted insertion of the CAR gene delivered to specific locations. It is a precise and safer method that kills cancer cells over time and improves T cell effectiveness [82], which benefits from endogenous T cell receptor, histocompatibility, and inhibitory receptor gene modification.

#### 3.2.5. In Vivo Delivery

Due to the restriction of many different barriers in vivo, developing the nanocarrier for CRISPR-Cas9 with safety and efficiency remains a challenge. DNA (in the form of a plasmid or virus) or mRNA delivery can cause Cas9 protein expression in targeted cells, resulting in Cas9-mediated gene editing [83]. For efficient in vivo delivery of CRISPR-Cas9, viral vectors such as adeno-associated virus (AAV) have been widely used in various studies [84,85]. However, the Cas9 gene is 4–7 kB and the amount of DNA that can be encapsulated in the virus is limited due to the maximum capacity of AAV being 4.7 kb [86]. Non-viral in vivo delivery of nucleases in direct mRNA or protein forms may be the best candidate vectors. Non-viral delivery, including nanoparticle-mediated delivery, may have some advantages over plasmid DNA or viral vectors for their stability and capacity. Various anti-cancer components, including chemotherapeutic drugs, have been developed and have demonstrated satisfactory results in inhibiting the growth of cancer cells using nanotechnology [44,87,88]. The biggest barrier to the clinical use of CRISPR-Cas9 is the lack of effective and safe delivery systems. Hence, the therapeutic CRISPR-Cas9 components must directly enter the target cells by overcoming various physical barriers in order to achieve effective and precise cancer treatment. Additionally, the functioning Cas9 protein and sgRNA must be transported to the nucleus simultaneously for the gene editing process [89]. The main focus of clinical application is the creation of nanocarriers to deliver CRISPR-Cas9 into specific cancer cells [90]. Polymer nanoparticles [88,91,92], lipid nanoparticles [93,94,95], and porous nanoparticle [96] are used as nanocarriers.

#### 3.2.6. Avoiding Off-Target Effect

Previous research has demonstrated that the CRISPR-Cas9 system frequently generates indels at undesirable genomic loci [97]. Continued genetic modification raises the risk of off-target cleavage and reduces editing specificity, potentially resulting in unwanted mutations and toxicity. Using paired Cas9 nickases, truncated gRNAs with shorter protospacer complementary regions, and high-fidelity Cas9 endonucleases could reduce the risk of Cas9-mediated off-target. Cas9 protein modification to change PAM preferences or improve target DNA recognition can also be used to reduce off-target effects and thus improve on-target specificity [98,99]. In addition, a synthetic switch was created to self-regulate Cas9 expression during both the transcription and translation steps. The synthetic switch could inhibit transcription and translation at the same time, rapidly reducing Cas9 expression. Cas9 expression was restricted to reduce off-target effects while increasing efficiency and on-target indel mutation [100]. The future CRISPR-Cas9 technology will not only reduce the possibility of off-target effects, but will also maintain on-target efficiency and specificity.

## 4. Therapeutic Approaches for Infectious Disease

Traditional virus defense includes vaccine prevention, i.e., antiviral treatment to prevent viral replication and symptom reduction. However, for some viruses, there are often no effective vaccines to prevent infection and, in some cases, antiviral treatment is also limited to preventing more severe symptoms [101]. Furthermore, numerous human viruses, including human immunodeficiency virus (HIV), herpes simplex virus (HSV), Epstein–Barr virus (EBV), human cytomegalovirus (HCMV), and Kaposi’s sarcoma-associated herpesvirus (KSHV), have been shown to cause latent infections [102]. Several major human viruses, such as hepatitis B virus (HBV), HSV, and HIV-1, exhibit chronic infection characteristics that involve stages of both silent and productive infection in persistent low-level viral replication [103]. Targeting the viral genome in latent and chronic infections is difficult because the genome is retained inside the host cell, either as free-floating viral minichromosomes or as incorporated into the host genome [104].

The CRISPR-Cas9 system presents a new defense technique against viral infection. It is used to study the virus–host interaction in greater detail and to develop a faster and more accurate diagnostic technique than the current system. Moreover, CRISPR-Cas9 technology can be used for gene editing in prevention and treatment by targeting the viral genome and the host genome itself to prevent virus proliferation or intrusion into the host. To be more specific, the principle of preventing viral infection using the CRISPR-Cas9 system has four major mechanisms [101] (Figure 3).

The modification of receptors for viral entry: Interactions between viral proteins and cell membrane receptors allow the virus to enter the host cell. In addition to interfering with viral tropism, CRISPR-Cas-induced editing of receptor genes can prevent virus–receptor binding and limit virus entry and spread. Modifying these receptors, which also aid in viral genome replication and packaging, can impede viral multiplication [105].The segmentation of host viral factors: For replication and propagation, the virus is primarily dependent on host proteins. Some of the genes that encode proteins essential to viruses can be silenced using the CRISPR-Cas-induced knock out, preventing viral replication [106].The induction of host transcriptional restriction factor: These factors are restricted by the coupling of inactive Cas9 and viral RNA, which blocks replication and leads to a reduction in viral RNA [107].The excision and deletion of integrated viral genome: Viral genes may be excised using CRISPR-Cas in cases where viruses integrate their DNA into the host genome via the deletion and inactivation of genes [108].

**Figure 3 bioengineering-09-00477-f003:**
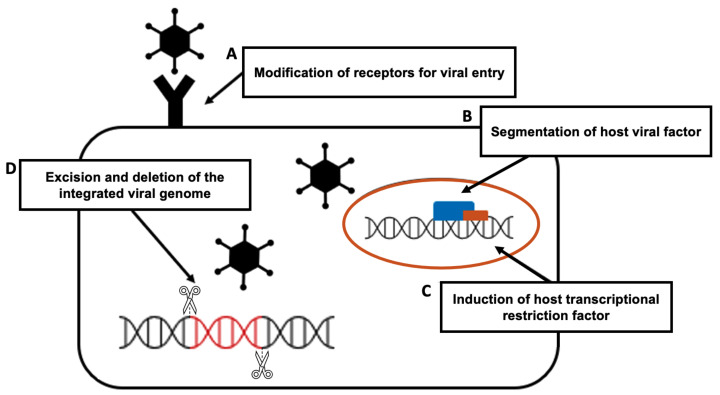
Mechanisms for preventing viral infection using the CRISPR-Cas system: (**A**) the modification of viral entry receptors by CRISPR-Cas-induced receptor gene editing; (**B**) the segmentation of host viral factors by the CRISPR-Cas-induced knockdown of proteins fundamental to viruses; (**C**) the induction of host transcriptional restriction by coupling inactive Cas and viral RNA; (**D**) the excision and deletion of integrated viral DNA by the CRISPR-Cas system.

### 4.1. HIV

HIV is an RNA virus that uses reverse transcription to produce the DNA required for replication and survival, and then intervenes in the host’s genome to cause latent infection. HIV attacks the infected host’s CD4+ T cells, resulting in acquired immune deficiency syndrome (AIDS) which neutralizes the host’s immune response, leaving the host vulnerable to secondary infections [109]. Highly active antiretroviral therapy (HAART), a combination of three or more drugs, significantly reduce HIV replication but cannot eradicate the provirus in resting CD4+ T cells and have no effect on latent infection [110].

(1)C-X-C chemokine receptor type 4 (CXCR4) and C-C chemokine receptor type 5 (CCR5) are co-receptors required for HIV-1 entry into CD4+ T cells. Therefore, HIV infection can be prevented by using CRISPR-Cas9 to create a defect in this receptor, and the disruption of genes encoding these co-receptors showed no obvious cytotoxic effects on cell viability, as well as a significant protective effect against HIV-1 infection when compared to unmodified cells [111].(2)To treat latent HIV infection, proviral DNA integrated in the host genome must be inactivated. CRISPR-Cas9 targeting of the long terminal region (LTR) enables this. The results of gene editing of the HIV-1 LTR U3 region revealed that Cas9/gRNAs completely excised a 9709 bp fragment of integrated proviral DNA spanning from 5 to 3 LTRs, resulting in viral gene expression inactivation and virus replication restriction in HIV-1 latently infected cells. CRISPR-Cas9-targeted proviral DNA has also been shown to prevent new HIV infections [112]. HIV-1 RNA editing with CRISPR-Cas13 is another effective treatment for HIV eradication. The findings suggest that the CRISPR-Cas13 system can effectively inhibit HIV-1 in primary CD4+ T-cells and reduce HIV-1 reactivation in latently infected cells [113].(3)The activation of restriction factor expression in host cells could be an alternate method for preventing HIV-1 replication. Restriction factors found recently, including the human silencing hub (HUSH) and NONO, are expected to effectively suppress HIV replication through reactivation using the CRIPSR-Cas9 [114,115].

### 4.2. Severe Acute Respiratory Syndrome Coronavirus 2 (SARS-CoV-2)

The pandemic of COVID-19 posed a serious threat to worldwide public health and economies. The first step in dealing with the emergence of such a fast-spreading novel virus is to understand the new virus’s characteristics and to precisely and quickly detect the infection. Following the discovery of the SARS-CoV-2, researchers used a wide CRISPR genome screen to identify that SARS-CoV-2 infiltrated the host cell surface via angiotensin-converting enzyme 2 (ACE2) [116]. For diagnositcs, the CRISPR system is used to construct a diagnostic assay that detects SARS-CoV-2 nucleic acid. A schematic representation of specific high-sensitivity enzymatic reporter unlocking (SHERLOCK), which is based on CRISPR-Cas13a, delivers a fast diagnostic and has been approved by the CDC in the United States. Since then, a simpler and faster diagnostic method based on CRISPR-Cas 12a called all-in-one dual CRISPR–Cas12a (termed “AIOD-CRISPR”) has been developed [117].

The prophylactic antiviral CRISPR in human cells (PAC-MAN), which is a CRISPR-Cas13-based technique, was found to be successful in targeting the highly conserved sequences of SARS-CoV-2. In human lung epithelial cells, the PAC-MAN method was shown to effectively degrade SARS-CoV-2 fragments. Then, additional analysis showed that a group of only six crRNAs can target more than 90% of corona viruses [118]. Another option is to use the CRISPR-Cas system to induce mutations in the host genome expressing ACE2 of the cell that binds when SARS-CoV-2 invades. This suggests the possibility of weakening SARS-CoV-2 binding by modifying the structure of the protein without losing the existing function by changing the existing ACE2-expressing genes [119].

### 4.3. Herpes Viruses

The EBV virus does not produce virus progeny during latent infection and only expresses a small number of viral proteins as well as non-coding RNAs, including several miRNAs. Further research revealed that CRISPR/Cas9 gRNAs targeting viral miRNAs such as BART5, BART6, or BART16 efficiently downstreamed miRNAs, and that the direct editing of the latent EBV genome in EBV-positive tumor cells was possible. Furthermore, data show that targeting the viral EBV nuclear antigen 1 (EBNA1) and various regions within the EBV origin of replication (OriP) with CRISPR-Cas9 significantly reduces viral genome content in latently infected EBV-infected cells [120].

With the exception of UL84, seven critical genes involved in the initiation of viral DNA replication, including UL54, UL44, UL57, UL70, UL105, UL86, and UL84, were targeted using CRISPR-Cas9 and successfully inhibited virus replication [120]. Another study created two CRISPR-Cas9 antiviral methods to target the UL122/123 gene, which is a key regulator of lytic replication and latency reactivation. The single-plex method targets the start codon with a single gRNA, resulting in a reduction in immediate early (IE) protein expression. The multiplex method employs three gRNAs to excise the entire UL122/123 gene, which excises the IE gene in 90% of viral genomes and thus inhibits IE protein expression [121].

For herpes simplex virus, the majority of gRNAs that targeted critical HSV-1 genes efficiently inhibited viral replication. A recent study demonstrated that CRISPR-Cas9, which targeted UL52 and UL29 genes of the HSV-1 primase–helicase complex, efficiently inhibit viral replication with no cytotoxic effect in vero cells [120,122]. Another study devised designated a HSV-1-erasing lentiviral particles (HELPs) model to prevent HSV-1 replication and to successfully inhibit the formation of herpetic stromal keratitis (HSK) in a mouse model [123].

## 5. Limitations and Challenges

Current CRISPR therapeutics still have limitations. First, the most serious concerns about genome editing therapy is the possibility of off-target mutagenesis. The level and duration of Cas9/gRNA expression also influences the risk [124]. Many efforts have been made to improve specificity, including the development of a high-priority sgRNA designer that incorporates multiple factors [125]. Second, on-target mutagenesis occurred frequently in double-strand breaks caused by single-guided RNA/Cas9, such as large deletions, over many kilobases and complex genomic rearrangements at the targeted sites, eliciting long-range transcriptional consequences and potentially pathogenic consequences [126]. The precise control of CRISPR-Cas9 activity in cells and complex conditions, such as cell-specific promoters, small molecule activation/inhibition, bioresponsive delivery carriers, and conditioned activation of the CRISPR-Cas9 system, will be beneficial [127]. Third, the efficient, safe, and targetable delivery of the CRISPR/Cas9 system in vivo is also a significant clinical challenge due to various physiologic barriers [128]. Fourth, another barrier to CRISPR/Cas9 application is the human body’s immunogenicity to the Cas9 protein derived from bacteria [129]. Fifth, the risk of repair events or genomic rearrangement following sgRNA-induced double-stranded breaks is also a concern in CRISPR-Cas9-based therapeutic interventions [130]. Although CRISPR-Cas9 technology can cause desired changes in genomic sequences, the poorly understood and less controlled DNA repair mechanism is associated with the risk of biological dysfunction [131]. Unexpected consequences of DNA repair mechanisms include the deletion of a few kilobases in the neighboring CRISPR-Cas9 nickase activity, the insertion (incorrect or partial) of donor DNA sequence to the site of integration, and inversion [132], which could lead to unexpected mutations [133]. However, CRISPR-Cas9 is still a developing technology that is used on patients with life-threatening conditions. Continuous efforts and clinical trials (summarized in Table 1) are being conducted to overcome the limitations of CRISPR-Cas therapeutics.

## 6. Conclusions

CRISPR/Cas9 has emerged and advanced rapidly as a stable, efficient, simple, and widely used gene-editing technology in just a few years. CRISPR/Cas9 has had a significant impact on many medical fields, including genetics, oncology, and infectious disease. Because the off-target effect of CRISPR/Cas9 gene-editing technology has always been a major concern, careful experimental design and thorough data analysis enabled therapeutic gene editing to produce useful results.

## Figures and Tables

**Figure 1 bioengineering-09-00477-f001:**
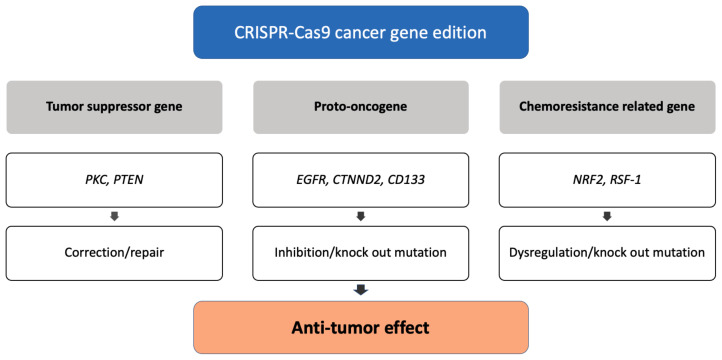
The mechanism of antitumor effect with CRISPR-Cas9 gene editing.

**Figure 2 bioengineering-09-00477-f002:**
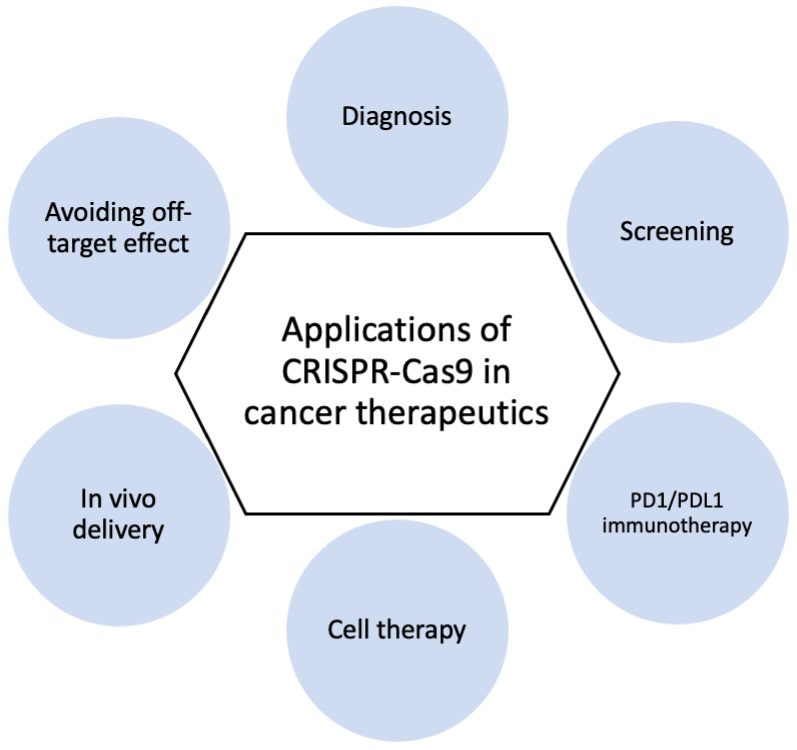
Applications of CRISPR-Cas9 in cancer therapy.

**Table 1 bioengineering-09-00477-t001:** Currently registered interventional clinical trials with CRISPR-Cas9-based gene editing.

NCT No.	Disease Type	Disease	Target	Intervention	Phase	Country
NCT03655678	genetic disease	β-thalassemia	disruption of the erythroid	ex vivo-modified hematopietic stem cell	I/II	USA
NCT04208529	genetic disease	β-thalassemia	disruption of the erythroid	ex vivo-modified hematopietic stem cell	I/II	USA
NCT03745287	genetic disease	sickle cell disease	disruption of the erythroid	ex vivo-modified hematopietic stem cell	I/II	USA
NCT04925206	genetic disease	β-thalassemia	disruption of the erythroid	ex vivo-modified hematopietic stem cell	I	China
NCT04774536	genetic disease	Sickle cell disease	disruption of the erythroid	ex vivo-modified hematopietic stem cell	I/II	USA
NCT03872479	genetic disease	Congenital Amaurosis	eliminate CEP290 mutation	gene editing product	I	USA
NCT04601051	genetic disease	Amyloidosis	disruption of the amyloid	Gene edit product in nanoparticle	I	UK, Swden
NCT04637763	cancer	B-cell lymphoma	creation of CD19-directed T cell	CAR-T cell to CD19	I	USA
NCT04035434	cancer	B-cell lymphoma	creation of CD19-directed T cell	CAR-T cell to CD19	I	USA
NCT05066165	cancer	Acute Myeloid Leukemia	create CD19-directed T cell	CAR-T cell to WT1	I	USA
NCT02793856	cancer	Non small cell lung cancer	PD-1 knock out	CAR-T cell with PD-1 knock out	I	China
NCT04842812	cancer	solid tumor	PD-1 knock out	CAR-T cell with PD-1 knock out	I	China
NCT04990557	Infectious disease	COVID-19	PD1 and ACE2 knockout	ex vivo-modified T cell	I/II	not specified

Search date: 6 September 2022; ClinicalTrials.gov.

## Data Availability

Not applicable.

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
