# Peer review of "Therapeutic Applications of the CRISPR-Cas System"

_bioengineering, 2022, doi:10.3390/bioengineering9090477_

Round 1

Reviewer 1 Report

Authors reviewed extensively from basic mechanism of CRISPR-Cas9 system to its clinical application in variable clinical fields. Corrections and additions are required in the following content.

1.  In the cancer therapeutics section, authors described genomic editing of cancer cells in cell lines. As described in other sections, it is recommended to add the contents of clinical application of CRISPR-Cas9 system in cancer diagnosis and treatment. (Zhang et al. Mol Cancer 2021: 20: 126)

2.     It is recommended to briefly describe the limitations and future directions of CRISPR-Cas9 technology (eg. off-target effect, editing efficiency and delivery methods) (Chen et al. Cancer letters 2019: 447: 48-55)

3. Please indicate the references cited faithfully.

4. Please check the spelling of page2 line71 (approachapproach) and page4 line 145 (-catenin).

Author Response

Respond to reviewer

  1. In the cancer therapeutics section, authors described genomic editing of cancer cells in cell lines. As described in other sections, it is recommended to add the contents of clinical application of CRISPR-Cas9 system in cancer diagnosis and treatment. (Zhang et al. Mol Cancer 2021: 20: 126)
  • Thank you for the important comment, we added cancer diagnosis and therapy section comprising CRISPR based diagnosis, CRISPR screening, PD1/PDL1 immunotherapy, cell therapy, In vivo delivery, and avoiding off-target effect

  1. It is recommended to briefly describe the limitations and future directions of CRISPR-Cas9 technology (eg. off-target effect, editing efficiency and delivery methods) (Chen et al. Cancer letters 2019: 447: 48-55)
  • Thank you for the important comment, we added Limitations and challenges including list of clinical trials
  1. Please indicate the references cited faithfully.
  • Thank you for the important comment, we added every references indicated.
  1. Please check the spelling of page2 line71 (approachapproach) and page4 line 145 (-catenin).
  • Thank you for the important comment. We corrected errors.

Reviewer 2 Report

The authors systematically reviewed the application of CRISPR-Cas9 gene editing system in the treatment of infectious diseases, genetic diseases and cancer diseases, and provided some theoretical and practical guidance in terms of therapeutic research of gene editing. Compared with other reviews which mostly focus on the classification, delivery methods and technical optimization of CRISPR-Cas-mediated gene editing, this paper provides a new perspective and certain novelty. However, the process and output of CRISPR-Cas9 are often complicated, researchers need to be careful about how to use it. The authors’ work is worthy of recognition, but I have some minor comments as follows:

1. in this review manuscript, the accuracy and safety of the application of Cas9 technology in each disease are rarely described, which is one of the most concerning issues in the clinical translation of Cas9 technology.

2. Many of the current clinical trials using the CRISPR platform are conducted using chimeric antigen receptor (CAR) T cells, and the authors might provide an appropriate overview of this technique.

3. The authors might consider adding a table at the end of the text that summarizes current clinical trials or preclinical studies of Cas9-based disease treatments.

4. Line 17-19: In the description of the bacterial CRISPR mechanism, the author described the process of intracellular CRISPR formation after the initial infection of DNA viruses. But in the process of the Type I system, the Cascade (CRISPR-associated complex for antiviral defence) would be formed, then Cas3 would cleave the target DNA. This paragraph only mentioned that Cas9 is involved in this process, rather than Cas3 or other elements of Cascade. It is suggested to give a further elaboration of the Type I mechanism.

5. Line 19: As with other abbreviations, please provide the full name of Cas9 on the first mention in this review.

6. Line 127: “but they can be used only for patients…”. It seems that there is a pronoun problem here.

7. Line 132: “the development of the CRISPR CAS system…”According to the entire manuscript, here should be changed to “CRISPR-Cas”.

8. Line 136: Furthermore, by using the the CRISPR…”. “the” appears twice. The authors should check the entire manuscript for spelling and grammatical errors.

9. Line 157: “acting as a tumor suppressor” might not agree in grammar and number with other words in this sentence. The suggestion is rewritten to “act as tumor suppressors”.

10. Line 221: CRISPR-Cas13 is an RNA editing system, but not a method, this sentence might be better if it is changed to “HIV-1 RNA editing with CRISPR-Cas13 is another effective treatment for HIV eradication”.

11. Line 275-280: In the conclusion, the author should discuss the future application prospect of CRISPR-Cas9 technology and the problems to be solved.

12. Figure 2: The schematic is relatively simple, does not fully describe the mechanism of the CRISPR-Cas9 system to prevent viral infection, and there is no figure legend. 2 and 3 should be separate, and add legend or annotate on the figure.

Author Response

  1. in this review manuscript, the accuracy and safety of the application of Cas9 technology in each disease are rarely described, which is one of the most concerning issues in the clinical translation of Cas9 technology.

- thank you for the important comment, we added sections of in vivo delivery, avoiding off-target effect and limitations & challenges

  1. Many of the current clinical trials using the CRISPR platform are conducted using chimeric antigen receptor (CAR) T cells, and the authors might provide an appropriate overview of this technique.

- thank you for the important comment, we added sections of cell therapy and the overview of CAR-T technique

  1. The authors might consider adding a table at the end of the text that summarizes current clinical trials or preclinical studies of Cas9-based disease treatments.

- Thank you for the important comment, we added a table of currently registered CRISPR based clinical trials

  1. Line 17-19: In the description of the bacterial CRISPR mechanism, the author described the process of intracellular CRISPR formation after the initial infection of DNA viruses. But in the process of the Type I system, the Cascade (CRISPR-associated complex for antiviral defence) would be formed, then Cas3 would cleave the target DNA. This paragraph only mentioned that Cas9 is involved in this process, rather than Cas3 or other elements of Cascade. It is suggested to give a further elaboration of the Type I mechanism.

- Thank you for the important comment, we added a mechanim (class and type) CRISPR-cas system

  1. Line 19: As with other abbreviations, please provide the full name of Cas9 on the first mention in this review.

- Thank you for the important comment, we added full name of Cas.

  1. Line 127:“but theycan be used only for patients…”. It seems that there is a pronoun problem here.

- Thank you for the important comment, we corrected pronoun errors.

  1. Line 132: “the development of the CRISPR CAS system…”According to the entire manuscript, here should be changed to “CRISPR-Cas”.

- Thank you for the important comment, we changed accordingly.

  1. Line 136: “Furthermore, by using thethe CRISPR…”. “the” appears twice. The authors should check the entire manuscript for spelling and grammatical errors.

- Thank you for the important comment, we changed accordingly.

  1. Line 157: “acting as a tumor suppressor” might not agree in grammar and number with other words in this sentence. The suggestion is rewritten to “act as tumor suppressors”.

- Thank you for the important comment, we changed accordingly.

  1. Line 221: CRISPR-Cas13 is an RNA editing system, but not a method, this sentence might be better if it is changed to “HIV-1 RNA editing with CRISPR-Cas13 is another effective treatment for HIV eradication”.

- Thank you for the important comment, we changed accordingly

  1. Line 275-280: In the conclusion, the author should discuss the future application prospect of CRISPR-Cas9 technology and the problems to be solved.

- Thank you for the important comment, we added limitations and challenges.

  1. Figure 2: The schematic is relatively simple, does not fully describe the mechanism of the CRISPR-Cas9 system to prevent viral infection, and there is no figure legend. 2 and 3 should be separate, and add legend or annotate on the figure.

- Thank you for the important comment, we modified all figures.

Round 2

Reviewer 1 Report

It's a well-written paper that's acceptable.

Reviewer 2 Report

I'm very pleased to see that the authors had greatly modified and refined the format and content of the manuscript. At present, in my opinion, this review can offer a certain academic reference for researchers of CRISPR-Cas9-based gene therapy. It is recommended that this manuscript is acceptable for publication in Bioengineering.